# Microcystin-LR in Primary Liver Cancers: An Overview

**DOI:** 10.3390/toxins14100715

**Published:** 2022-10-20

**Authors:** Shen Gu, Mingxuemei Jiang, Bo Zhang

**Affiliations:** 1Key Laboratory of Clinical Cancer Pharmacology and Toxicology Research of Zhejiang Province, Affiliated Hangzhou First People’s Hospital, Zhejiang University School of Medicine, Hangzhou 310006, China; 2Translational Medicine Research Center, Affiliated Hangzhou First People’s Hospital, Zhejiang University School of Medicine, Hangzhou 310006, China; 3Institute of Scientific and Technical Information of Zhejiang Province, Hangzhou 310001, China

**Keywords:** microcystin-LR, primary liver cancers, tumor origin, tumor progression

## Abstract

The cyanobacterial blooms produced by eutrophic water bodies have become a serious environmental issue around the world. After cellular lysing or algaecide treatment, microcystins (MCs), which are regarded as the most frequently encountered cyanobacterial toxins in fresh water, are released into water. Among all the variants of MCs, MC-LR has been widely studied due to its severe hepatotoxicity. Since 1992, various studies have identified the important roles of MC-LR in the origin and progression of primary liver cancers (PLCs), although few reviews have focused on it. Therefore, this review aims to summarize the major achievements and shortcomings observed in the past few years. Based on the available literature, the mechanisms of how MC-LR induces or promotes PLCs are elucidated in this review. This review aims to enhance our understanding of the role that MC-LR plays in PLCs and provides a rational approach for future applications.

## 1. Introduction

Contamination with harmful cyanobacterial blooms has become a serious environmental issue around the world [1]. Cyantoxins are essentially endotoxins that are released in water following cellular lysing or treatment with algaecides [2,3]. All major orders of *Cyanobacteria* can produce a family of hepatotoxins called microcystins (MCs), which are the most frequently encountered cyanotoxins in fresh water [4,5]. More than 279 congeners of MCs have been identified with molecular weights in the range of 882–1101 Da [6,7]. MC-LR—MC combined with leucine (L) and arginine (R) at positions 2 and 4 (Figure 1)—is the most studied congener due to its ubiquity and toxicity [8,9]. The unique cyclic structure of MC-LR protects it from oxidation, heat, and hydrolysis, enhancing its stability in the environment and its resistance to biodegradation [10,11,12]. MC-LR can pose a threat to animals and humans through contaminated drinking water or through the food chain (Figure 2).

MC-LR can easily enter cells via organic anion transporting polypeptides (OATPs) and accumulate in the target organs through blood circulation. As such, MC-LR can damage almost every system in the body, including the digestive system (liver [13,14], stomach [15], intestines [16], and pancreas [15]), nervous system (brain [17,18]), respiratory system (lungs [19,20,21]), circulatory system (heart [22]), dermal system (skin [22,23]), genital system (testicles [24], prostate [25], ovaries [26]), etc. In addition, experimental studies have indicated that MC-LR exposure may play an important role in the origin and progression of various cancers [27,28,29]. Thus, MC-LR was classified as a Group 2B carcinogen by the International Agency for Research on Cancer in 2010 [30].

Primary liver cancer (PLC) ranks as the sixth most common cancer and the third leading cause of cancer-related death in the world, accounting for more than 800,000 deaths in 2020 [31]. The main types of PLC include hepatocellular carcinoma (HCC) and intrahepatic cholangiocarcinoma (ICC), alongside other rare types [32]. HCC and ICC are both malignant tumors originating from hepatocytes and the biliary epithelium, respectively [33]. The biological behaviors of HCC and ICC are completely different. MC-LR was first reported as a potent liver cancer promoter in 1992, and a range of related studies has been published in last 30 years [14]. However, there are few reviews focusing on the role of MC-LR in PLC. Previous studies focused mainly on the relationship between HCC and MC-LR, while recent research has discussed the role of MC-LR in ICC [34]. For a deeper understanding of the effects of MC-LR in PLCs, data on the function and mechanism of MC-LR in PLC were collected in this paper and these findings were reviewed. 

## 2. Liver Fibrosis and Cirrhosis induced by Microcystin-LR

Most chronic liver diseases, including hepatocellular carcinoma and intrahepatic cholangiocarcinoma, have been confirmed to be related to liver fibrosis [35,36]. Gu et al. confirmed that exposure to MC-LR at 15 μg·kg^−1^ could induce liver fibrosis in mice. Furthermore, MC-LR was found to be able to enter and activate hepatic stellate cells, resulting in differentiation and impacting Hedgehog signaling in cells [35]. However, Meaghan et al. reported that chronic treatment with MC-LR would have no effect on hepatic stellate cell metabolism, proliferation, or activation, and that it was unlikely to lead to chronic liver disease [37].

Nonalcoholic steatohepatitis (NASH) causes extracellular matrix remodeling in the liver and is a risk factor for HCC. MC-LR has been shown to increase lipid accumulation in the liver tissues [38,39,40]. He et al. identified that MC-LR could significantly inhibit fatty acid β-oxidation and promote hepatic inflammation, resulting in NASH [41]. Using a histological staining method, it was shown [42] that, besides the induction of NASH, MC-LR could aggravate NASH induced by a high-fat or high-cholesterol (HFHC) diet.

## 3. Microcystin-LR in Hepatocellular Carcinoma

Hepatocellular carcinoma (HCC) is the most common type of primary liver cancer, comprising 75%–85% of cases, with China being one of the most high-risk areas for HCC in the world [31]. Early studies reported that microcystin-LR (MC-LR) dose-dependently increased the number of neoplastic foci in rat liver, which was initiated with diethyl nitrosamine [14]. Additionally, more and more research has proven that MC-LR is closely related to the origin and progression of HCC through in vitro and in vivo studies.

### 3.1. MC-LR in the Origin of HCC

Clinical and experimental studies both show that MC-LR plays a significant role in the incidence of HCC. This section summarizes the role of MC-LR in the origin of HCC in experimental and clinical studies. Experimental studies have revealed the effects of MC-LR on hepatocytes and the liver. Clinical and epidemiological studies include research into the association between freshwater/serum MC-LR content and the incidence of HCC. 

#### 3.1.1. Experimental Research

##### MC-LR Acts as an Inhibitor of Phosphatases 2A in Hepatocytes

Phosphatase 2A (PP2A)—one of the major cellular Ser/Thr protein phosphatases in cells—is widely considered as a tumor suppressor due to its function in DNA repair, autophagy, proliferation, differentiation, and apoptosis [43,44,45]. In 1990, Carol et al. first reported that MC-LR could inhibit PP2A isolated from both higher plants and mammals with *K_i_* values below 0.1 nM. The hyper-phosphorylation of cytokeratins is closely associated with liver tumor promotion [46], and treatment with MC-LR in rat hepatocytes has been proven to induce the hyper-phosphorylation of cytokeratins 8 and 18 by inhibiting the PP2A activity in cells [47]. Abnormal hepatocyte proliferation related to the inhibition of PP2A activity was also detected after MC-LR exposure during in vitro and in vivo studies. Liu et al. identified that MC-LR could inhibit PP2A activity and hyper-phosphorylate Akt, S6K1, S6, and 4E-BP1 in HL7702 cells, cells from a normal human liver, resulting in cell proliferation in vitro. For in vivo studies, Liu et al. also proved that 80 mg·kg^−1^ MC-LR could promote abnormal liver cell proliferation beginning at 1 d post exposure, and that proteins related to the Akt and MAPK signaling pathways were hyper-phosphorylated in mice liver via the inhibition of PP2A activity [48,49]. 

##### MC-LR Induces Oxidative Stress in Hepatocytes

Oxidative stress has been recognized as a liver cancer-initiating stress response [50]. MC-LR can induce the accumulation of reactive oxygen species (ROS) in hepatocytes, contributing to the formation of hepatocellular carcinoma (HCC) [51]. Due to its human origin and because it retains the specialized liver functions present in human hepatocytes, the HepG2 cell line was often regarded as a substitute for liver cells in vitro studies [52]. It has been experimentally shown [53] that chronic exposure (for 83 days) to MC-LR can increase the level of ROS in HepG2 cells. Cytochrome P450 2E1 (CYP2E1) is an important contributor to MC-LR-induced oxidative stress [54], and Sun et al. discovered that in mouse liver receiving short-term MC-LR treatment at 40 μg·kg^−1^, the mRNA and protein levels of CYP2E1 were obviously increased, indicating that acute MC-LR exposure can induce the accumulation of ROS in the liver [55]. These findings suggest that ROS may play a significant role in the tumor-promoting mechanisms of MC-LR. 

Oxidative DNA damage is confirmed to play a role in the development of some cancers [56,57,58]. Oxidative DNA damage-specific enzymes such as formamidopyrimidine-DNA glycosylase and endonuclease III were found to be markedly increased in MC-LR-treated HepG2 cells, providing evidence that MC-LR can induce DNA strand breaks in liver cells [59]. MC-LR was also proven to enhance 8-oxo-dG in the rat liver and in primary cultured hepatocytes, while 8-oxo-dG was the most frequent biomarker for oxidative DNA damage induced by ROS [60].

##### MC-LR Influences Tumor-Associated Non-Coding RNAs in Hepatocytes

According to their location, shape, and length, non-coding RNAs (ncRNAs) are divided into different classes. Among them, microRNA (miRNA), circular RNA (circRNA), and long ncRNA (lncRNA) are the three major types of ncRNA with distinct functions in cancers [61]. 

MiRNAs are small single-stranded RNAs containing an average of 22 nucleotides. MiRNAs can downregulate the expression of their target genes by binding to recognition sequences, resulting in miRNA-mediated target gene degradation and translational decay or repression [62,63]. An analysis of miRNA expression profiling was conducted in HepG2 cells treated with MC-LR and the expression of has-miR-454-3p, has-miR-449c-5p and has-miR-149-3p was increased in MC-LR-treated cells, while the expression of has-miR-500b-5p, has-miR-500a-5p, has-miR-500a-3p, and has-miR-4286 was decreased [64]. After exposure to MC-LR, changed miRNA expression was also detected in the liver of juvenile silver carp using small-RNA sequencing, with 372 miRNAs significantly changed after 24 h of treatment with MC-LR [65]. MiRNA profiling of HL7702 cells exposed to MC-LR was determined via high-throughput sequencing techniques. Compared to the control cells, the expression levels of miR-4521 were significantly reduced in all treatment groups, while the expression levels of miR-15b-3p were markedly increased. Kyoto Encyclopedia of Genes and Genomes (KEGG) pathway enrichment analysis data showed that the target genes of differentially expressed miRNAs predominantly participated in cancer development [66]. Some studies have pointed out that the expression of certain miRNAs play an important role in the pathogenesis of HCC [67,68]. Thus, for early HCC detection, Xu et al. established a liver cancer model in mice using low doses of MC-LR and detected the miRNA changes in the experimental groups. The detection of seven miRNAs may prove to be an effective method for the prediction of hepatocarcinogenesis caused by MC-LR [69].

Circular RNAs (circRNAs) are generated by back-spliced exons at the pre-mRNA stage to form covalently closed continuous loop structures without a 5ʹ or 3ʹ polarity structure [70,71]. CircRNAs bind to miRNA, acting as a “sponge” to regulate the expression of the genes encoding proteins at the posttranscriptional and transcriptional levels [72]. Using high-throughput sequencing analysis, Zheng et al. demonstrated the expression levels of the circRNAs in HL7702 cells after exposure to MC-LR at concentrations ranging from 1–10 μM. The findings showed that the expression levels of hsa_circRNA_0000659 and hsa_circRNA_0000657 were downregulated while those of hsa_circRNA_0001535 and hsa_circRNA_0003247 were upregulated in all of the MC-LR-exposed groups [73]. Under the same exposure conditions, Wen et al. demonstrated the expression levels of lncRNAs in HL7702 cells after MC-LR exposure. The expression levels of LNC_00027, MIR22HG, and LINC00847 were markedly increased in all treatment groups [74]. The KEGG pathway enrichment analysis confirmed that MC-LR exposure activated some important signaling pathways in HCC, such as the MAPK and PI3K-Akt signaling pathway, through influencing the expression levels of particular circRNAs and lncRNAs [73,74]. 

##### MC-LR Induces DNA Methylation in Hepatocytes

Epigenetic alterations, including DNA methylation, were proven to be involved in the development and progression of hepatocarcinoma. The methylation of promoter CpG islands can inhibit gene transcription and cause the silencing of tumor suppressor genes as well as mutations and deletions [75,76]. Chen et al. detected CpG island methylation in the human hepatocyte L02 cell line treated with MC-LR and discovered that 2592 CpG sites differentially methylated in the promoter or the coding DNA sequence (CDS) of genes through DNA methylation sequencing. KEGG pathway analysis showed that significantly changed mRNAs were mainly involved in cancer formation, invasion, and migration [77]. In addition to the bioinformatics analysis above, Zhao et al. also proposed that MC-LR could downregulate the expression of Aristaless-like Homeobox 4 in the L02 cell line and in rat models via the promotion of methylation, resulting in HCC [78].

#### 3.1.2. Clinical and Epidemiological Research

Early clinical studies mainly focused on the relationship between HCC and the quality of drinking water [79]. These data indicated that people who drank water contaminated with microcystins had a higher HCC mortality rate compared to people who drank good quality water. Furthermore, differences have been observed between the quantity of MCs found in the drinking water of HCC patients and that in the water ingested by controls [80]. Similar results were also demonstrated in Serbia. Svircev et al. identified that a higher PLC occurrence might be associated with the presence of cyanotoxins in drinking water [81].

Recently, Zheng et al. conducted a clinical case-control study to investigate the relationship between serum MC-LR and HCC risk. Blood samples were collected from HCC patients and control subjects and were analyzed for serum MC-LR using ELISA. In this research, serum MC-LR was confirmed to be an independent risk factor for HCC in humans, with a negative interaction with aflatoxin and an obvious positive interaction with the hepatitis B virus or alcohol [82].

### 3.2. MC-LR in the Prognosis of HCC

Although early research from 1992 discovered that MC-LR participated in diethyl-nitrosamine-induced HCC [14], few studies focused on the poorer prognosis of HCC induced by MC-LR. Gan et al. found that MC-LR could promote liver cancer cell growth via the activation of nuclear factor erythroid 2-related factor 2 in HCC cell lines (HepG2 and Hep3B Cells) [83]. Lei et al. also confirmed the correlation between MC-LR and the prognosis of HCC patients in a clinical study. After the detection of MC-LR in the serum of patients with HCC, multifactorial *COX* regression analysis was conducted, and the results showed that a serum MC-LR concentration of ≥0.97 ng·mL^−1^ was related to an increased risk of tumor relapse. Thus, serum MC-LR could worsen prognosis in patients with HCC [84]. 

## 4. Microcystin-LR in Intrahepatic Cholangiocarcinoma

Intrahepatic cholangiocarcinoma (ICC) is a rare, highly aggressive, and often fatal primary epithelial cancer arising from the intrahepatic bile duct. Most ICCs develop in non-cirrhotic livers, and mass-forming ICCs are typically characterized by a hypo-vascularized tumor stroma and prominent desmoplastic stroma, making it different from conventional HCCs [85,86]. Recently, several investigations have discussed the function of MC-LR in the origin and progression of ICCs; these results are discussed below.

### 4.1. MC-LR in the Origin of ICC

In the past few years, the special effects of MC-LR on ICC have been demonstrated, and all these results have indicated that MC-LR might participate in the onset of ICC. Yan et al. conducted a study to investigate the toxic effects of MC-LR on intrahepatic biliary epithelial cells in vivo and in vitro. After chronic exposure, MC-LR-treated mice exhibited an obviously thickened bile duct morphology and bile duct hyperplasia. MC-LR can activate the ERK-RSK signaling in human primary intrahepatic biliary epithelial cells (HiBECs) and promote cell proliferation via inhibiting PP2A activity [87]. 

In addition to the direct effects on HiBECs induced by MC-LR, Yan et al. also focused on the interaction between HiBECs and surrounding cells after exposure to MC-LR in vivo and in vitro studies. A growing number of macrophages were evaluated as being in the portal area after exposure to MC-LR. To explore the molecular mechanism involved in this progression, a co-culture system including THP-1 cells and HiBECs was implemented in the presence of MC-LR. During exposure to MC-LR, the HiBECs had a large chemotactic effect on THP-1 cells and induced M2-type polarization. In turn, inflammatory factors in the medium of polarized THP-1 cells can induce the abnormal proliferation and migration of HiBECs via the activation of related pathways in HiBECs [88]. These results indicate that the interaction between HiBECs and the surrounding macrophages after exposure to MC-LR might promote the formation of ICC in humans.

### 4.2. MC-LR in the Prognosis of ICC

The prognosis for ICC remains dismal, despite the development of advanced therapeutic tools [89]. Previous studies have confirmed that the desmoplastic reaction in ICC plays an active and crucial role in promoting progressive and invasive ICC growth and metastasis [90,91]. Gu et al. conducted a retrospective study to evaluate the prognostic value of MC-LR in ICC after resection, and multivariate analysis showed that a high-MC-LR level in tumor issue was the independent prognostic factor for over-survival and recurrence-free survival after hepatectomy. In addition, MC-LR was proven to promote the survival of human ICC cell lines, and SET was identified to play an important role in this progression [34].

## 5. Summary

Microcystin-LR (MC-LR) has been shown to participate in the origin and progression of hepatocellular carcinoma (HCC) and intrahepatic cholangiocarcinoma (ICC) (Figure 3). MC-LR was found to be able to inhibit the activity of phosphatase 2A (PP2A) activity and to be able to activate the Akt and MAPK signaling pathways, resulting in abnormal cell proliferation in hepatocytes. The inhibition of PP2A in hepatic stellate cells induced by MC-LR was also found to promote liver fibrosis and to result in the development of HCC. MC-LR can induce the accumulation of reactive oxygen species and damage hepatocyte DNA, which have been confirmed to be related to the formation of HCC. In addition, MC-LR can regulate the expression of the proteins associated with hepatocarcinogenesis by influencing non-coding RNAs, including microRNAs, circular RNAs, long ncRNAs, as well as by influencing DNA methylation. Furthermore, some studies have demonstrated persistent carcinogenic changes and impaired hepatic recovery after MC-LR toxicity in nonalcoholic steatohepatitis—a risk factor for liver cirrhosis and HCC. Moreover, serum MC-LR levels have been confirmed to be involved in the onset and prognosis of HCC. Moreover, we have also proposed for the first time that MC-LR can induce abnormal cell proliferation in human intrahepatic biliary epithelial cells in vivo and in vitro via the direct inhibition of PP2A in biliary epithelial cells and that it has an indirect influence on the surrounding macrophages. Clinical studies have also been conducted to investigate the role of MC-LR in the poor prognosis of ICC patients, and MC-LR was identified as the independent prognostic factor for over-survival and recurrence-free survival.

## 6. Future Directions

The role of microcystin-LR (MC-LR) in primary liver cancers (PLCs) has been studied extensively, but there are still many problems to be solved. Firstly, the effects of MC-LR on the prognosis of hepatocellular carcinoma (HCC) have not been entirely confirmed. Although Lei et al. discovered that MC-LR can worsen the prognosis of HCC patients, patients who were not positive for chronic hepatitis B virus infection were not included, and those who were heavy drinkers were also not included in the study, severely reducing the reliability of the research. The molecular mechanism of MC-LR that is involved in the prognosis of HCC has not yet been explained with clarity and carefulness. Secondly, the scope of future studies should be expanded to include an in-depth focus on the role of MC-LR in intrahepatic cholangiocarcinoma (ICC), with the second incidence in PLCs. Previous studies have identified that MC-LR is associated with the origin and prognosis of ICC. However, the influence of MC-LR on the incidence of ICC in humans has not been confirmed. Past clinical studies have focused mainly on the relationship between MC-LR and HCC. In addition, how the mechanism of MC-LR induces pathological alteration in intrahepatic bile ducts and causes physiopathology changes still requires further study. A better understanding of MC-LR in PLCs may accelerate the progress of novel therapeutics to prevent liver cancer and may improve patient prognosis.

## Figures and Tables

**Figure 1 toxins-14-00715-f001:**
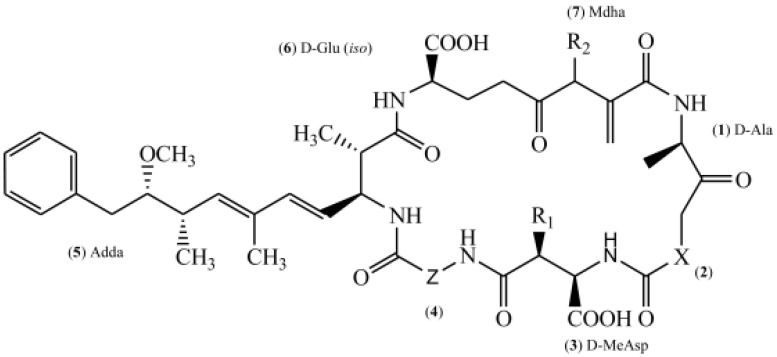
Chemical structure of microcystin. (X and Z represent different amino acids.) Abbreviation: Ala: Alanine, MeAsp: methylaspartic acid, Adda: 3-amino-9-methoxy-2, 6, 8-trimethyl-10-phenyldeca-4, 6-dienoic acid, Glu: glutamic acid, Mdha: methyl dehydroalanine.

**Figure 2 toxins-14-00715-f002:**
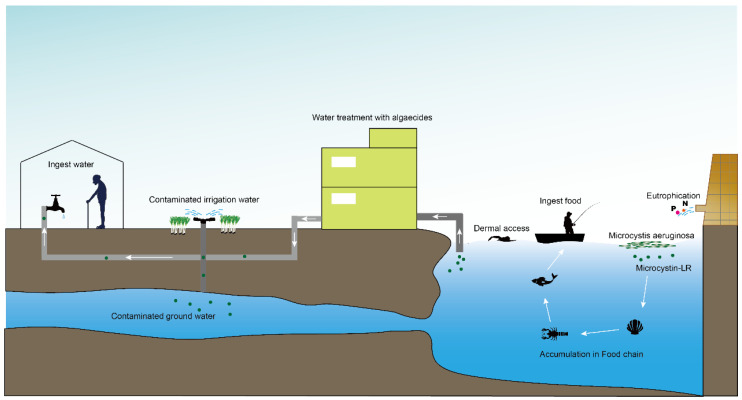
Scenarios of human and animal exposure to microcystin-LR in freshwater environments that are affected by cyanobacterial blooms: inhalation (recreation, agriculture, bathing, and other occupational activities), dermal (recreation, bathing, and so on), and ingestion (water and food). Abbreviation: P: phosphorus, N: nitrogen.

**Figure 3 toxins-14-00715-f003:**
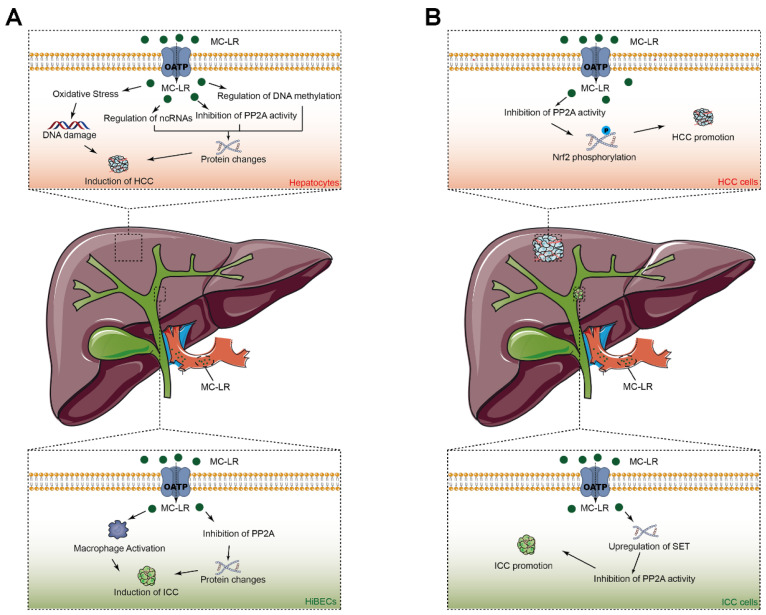
Schematic diagram of the possible molecular mechanisms underlying microcystin-LR-induced carcinogenesis (**A**) and tumor progression (**B**) of primary liver cancers. Abbreviation: HCC: hepatocellular carcinoma, ICC: intrahepatic cholangiocarcinoma, OATP: organic anion transport polypeptide, PP2A: protein phosphatase 2 A.

## Data Availability

No data was generated or used during the study.

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
