# Peer review of "Microcystin-LR in Primary Liver Cancers: An Overview"

_toxins, 2022, doi:10.3390/toxins14100715_

Round 1
Reviewer 1 Report
Dear Authors!
The paper is devoted to review of the published works on role of MC-LR in the origin and progression of primary liver cancers. I consider it important and useful work to bring together all the findings in this scientific field. Many documents have been cited. But in most cases there is no analysis of the collected information, only a listing of data.
The information is presented too wordy, some parts should be shortened. Below I have given examples. Unfortunately, some phrases are not clear. I see that the language correction has been made, but the meaning of the sentences was sometimes lost.
There are many inaccuracies in the introduction of the article. Refer to the document that I suggested you look at.
A few more comments below:
Lines 23. Microcystis aeruginosa is one among the range of microcystin-producing species.
I cite the Reference: Cyanobacterial toxins: microcystins. Background document for development of WHO Guidelines for drinking-water quality and Guidelines for safe recreational water environments. Geneva: World Health Organization; 2020 (WHO/HEP/ECH/WSH/2020.6): “MC-producing strains can be found in all major orders of Cyanobacteria – that is, Chroococcales, Oscillatoriales, Nostocales and Stigonematales. Microcystis is the most widely occurring genus with MC-producing species; other widespread ones include Anabaena (some species of which are now classified as Dolichospermum), Nostoc and Planktothrix.”
Line 25-26 “More than 279 congeners of MCs have been identified, with different amino acids in positions X and Z” - Not only due to these variations. “To date, more than 250 different MCs have been identified, with molecular weights in the range of 882–1101 Da (Spoof & Catherine, 2017; Bouaïcha et al., 2019). Structural variations exist in all seven amino acids; the most frequent are substitution of L-amino acids at positions 2 and 4, substitution of Mdha by dehydrobutyrine (Dhb) or serine at position 7, and a lack of methylation of amino acids at positions 3 and/or 7. The structural variations observed in Adda, although not frequent, can be of relevance, since they may affect analytical tests using Adda as a marker. The principle nomenclature of MCs is based on the variable L-amino acids at positions 2 and 4; for example, using the standard one-letter codes for amino acids, MC-LR contains L-leucine (L) at position 2 and L-arginine (R) at position 4. All other variations in the molecule are suffixed to the respective variant; for example, [D-Asp3 ]MC-LR lacks the methyl group at position 3.” I cited the above reference
Line 26 “MC-LR ̶ MCs combined with leucine” Between MC-LR and MCs it is necessary to put a dash here, not a hyphen. These are two different words meaning the one substance and the group of substances.
Line 26. In this sentence “ is the most studied genus due to its ubiquity and toxicity” I would exchange word “genus” to “ representative or congener”.
Line 29. “MC-LR is mainly present in water environments…” it is incorrect. The profile of MCs depends on the MC-producing species.
Lines 32 and further. Are you sure that it is true only for MC-LR, but not for all MCs? It is necessary to check.
Line 46-47. I would move the sentence “Previous studies mainly focused on the relationship between HCC and MC-LR, while recent research has discussed the role of MC-LR in ICC [32]”. After the sentence ”However, there are few reviews focusing on the role of MC-LR in PLC. (line 50)”
Line 60 There is no reference to Figure 3 in the text.
Is it your own picture (Figure 3) or taken from somewhere? And the same question for the picture 2. The references should be here if they are not yours.
Line 99-105 Some sentences should be rewritten in more plain manner. For example: “After MC-LR was exposed to HepG2 cells for 83 d, the ROS level in the MC-LR-treated cells was significantly higher than that of the control cells, which proved that chronic exposure to MC-LR could increase ROS levels in HepG2 cells [43]”. It could be: ” It has been experimentally shown [43] that chronic exposure (for 83 days) to MC-LR can increase the level of ROS in HepG2 cells.”
Lines 137-138 “The combination of seven miRNAs may prove to be an effective method for the detection of hepatocarcinogenesis caused by MC- LR [59]”. The sentence is unclear. It should be rewritten. MiRNAs are a small single-stranded non-coding RNA molecules and could not be a method. They can be used as a target compounds, can be detected for prediction of hepatocarcinogenesis caused by MC- LR.
Line 169 “the vehicle-exposed HFHC group” is it “placebo” group?
“Besides the induction of NASH, MC-LR could aggravate NASH induced by a high fat or high cholesterol (HFHC) diet. Histological staining suggested that the MC-LR-exposed HFHC group had more fibrosis and less steatosis compared to the MC-LR-exposed control group 168 and the vehicle-exposed HFHC group [68].” Could be just one sentence: “Using the method of histological staining, it was shown [68], that besides the induction of NASH, MC-LR could aggravate NASH induced by a high fat or high cholesterol (HFHC) diet”.
Line 185 “pond-ditch water, which contains various toxins, including MC-LR” could be “water contaminated with microcystins”
Line 187 “well or deep-well water’”. Do you mean “good quality” or “microcystin-free water”?
Line 187-189 The sentence is unclear: “Furthermore, differences have been observed in the quantity of MCs found in the drinking water of HCC patients and the in the water ingested by controls [74].
Line 190 Wrong citing “Zorica at al” Zorica is first name, Svircev is surname. (it is reference â„–75).
Please, follow the rule: “In the text, reference numbers should be placed in square brackets [ ], and placed before the punctuation.” Check, please, the order of brackets and punctuation. Links in the text should be referenced by numbers
Lines 213-214 “Though the relationship between HCC and microcystin-LR (MC-LR) has been widely investigated in the past 30 years, few researchers have paid attention to the role of MC-LR in ICC”. I would remove this sentence. It was in Introduction
Reviewer 2 Report
I've read with pleasure this well written Review about the role of Microcystin-LR in primary liver cancers. I have only one suggestion regarding its organization. I would discuss the role of oxidative stress and fibrosis-cirrhosis in two independent paragraphs before explaining in details the role of Microcystin-LR in HCC and ICC, since the two pathogenetic processes can share important features, which are significant for both cancer types.
Please check some minor typos throughout the text.
Reviewer 3 Report
This is a well written review summarizing the link between MCLR exposure and primary liver cancer. A few corrections:
- the abbreviation for primary liver cancer, PLC is sometimes interchanged to PCL throughout the manuscript.
- Check manuscript for spelling
- Line 98 - in vitro instead of vitro
Lines 167-169: MCLR has been shown to increase lipid accumulation in the liver tissues (Ref: Sedan, D., Laguens, M., Copparoni, G., Aranda, J.O., Giannuzzi, L., Marra, C.A. and Andrinolo, D., 2015. Hepatic and intestine alterations in mice after prolonged exposure to low oral doses of Microcystin-LR. Toxicon, 104, pp.26-33.;
Lad, A., Su, R.C., Breidenbach, J.D., Stemmer, P.M., Carruthers, N.J., Sanchez, N.K., Khalaf, F.K., Zhang, S., Kleinhenz, A.L., Dube, P. and Mohammed, C.J., 2019. Chronic low dose oral exposure to microcystin-LR exacerbates hepatic injury in a murine model of non-alcoholic fatty liver disease. Toxins, 11(9), p.486.;
Lad, A., Hunyadi, J., Connolly, J., Breidenbach, J.D., Khalaf, F.K., Dube, P., Zhang, S., Kleinhenz, A.L., Baliu-Rodriguez, D., Isailovic, D. and Hinds Jr, T.D., 2022. Antioxidant Therapy Significantly Attenuates Hepatotoxicity following Low Dose Exposure to Microcystin-LR in a Murine Model of Diet-Induced Non-Alcoholic Fatty Liver Disease. Antioxidants, 11(8), p.1625.).
- Some abbreviations have been explained repeatedly. Mention only the first time.
Round 2
Reviewer 1 Report
Dear authors!
thanks for the proper fixes.
There is only one thing left. I still haven't found a link to Figure 3 in the text.
Good luck.